

# The Impact of Altering Emission Data Precision on Compression Efficiency and Accuracy of Simulations of the Community Multiscale Air Quality Model

Michael S. Walters[§,†,★] and David C. Wong[§,★]

[§]Atmospheric and Environmental Systems Modeling Division, Center for Environmental Measurement and Modeling, Office of Research and Development, U.S. Environmental Protection Agency, Research Triangle Park, NC, USA
[†] Oak Ridge Associated Universities, Oak Ridge, TN, USA.
[★]These authors contributed equally to this work.

*Correspondence to:* David C. Wong (wong.david-c@epa.gov)

**Abstract.** The Community Multiscale Air Quality Model (CMAQ) has been a vital tool for air quality research and management at the United States Environmental Protection Agency (U.S. EPA), and at government environmental agencies and academic institutions worldwide. CMAQ requires a significant amount of disk space to store and archive input and output. For example, an annual simulation over the contiguous United States with horizontal grid cell spacing of 12 km requires 2−3

TB of input data and can produce anywhere from 7−45 TB of output data, depending upon modelling configuration, and desired post-processing output (e.g., for evaluations or graphics). After a simulation is complete, model data are archived for several years, or even decades, to ensure the replicability of conducted research. As a result, careful disk space management is essential to optimize resources and ensure the uninterrupted progress of ongoing research and applications requiring large scale, air quality modelling. Proper disk space management may include applying optimal data compression techniques that

are executed on input and output files for all CMAQ simulations. There are several (not limited to) such utilities that compress files using losslessness compression, such as GNU Gzip and Basic Leucine Zipper Domain (bzip2). A new approach is proposed in this study that reduces the precision of the air quality model emissions input to reduce storage requirements (after a losslessness compression utility is applied) and accelerate runtime. The new approach is tested using CMAQ simulations and post-processed CMAQ output to examine the impact on the air quality model performance. In total, four simulations were

conducted, and nine cases were post-processed from direct simulation output to determine disk space efficiency, runtime efficiency, and model (predictive) accuracy. Three simulations were run with emissions input containing only five, four, or three significant digits. To enhance the analysis of disk space efficiency, altered emissions CMAQ simulations were additionally post-processed to contain five, four, or three significant digits. The fourth, and final, simulation was run using the full precision emissions files with no alteration. Thus, in total, 13 gridded products (four simulations and nine cases) were

analysed in this study.





Results demonstrate that the altered emission files reduced the disk space footprint by 6 %, 25 %, and 48 % compared to the unaltered emission files when using the bzip2 compression utility for files containing five, four, or three significant digits, respectively. Similarly, the altered output files reduced the required disk space by 19 %, 47 %, and 69 % compared to the unaltered CMAQ output files when using the bzip2 compression utility for files containing five, four, or three significant digits, respectively. For both compressed datasets, bzip2 performed better than gzip, in terms of compression size, by 5−27 % for emission data and 15−28 % for CMAQ output for files containing five, four, or three significant digits. Additionally, CMAQ runtime was reduced by 2−7 % for simulations using emission files with reduced precision data on a non-dedicated environment. Finally, the model estimated pollutant concentrations from the four simulations were compared to observed data from the U.S. EPA Air Quality System (AQS) and the Ammonia Monitoring Network (AMON). Model performance statistics were negligibly impacted (e.g., normalized mean bias differed by less than 0.01 % for all altered simulations and cases). In summary, by reducing the precision of CMAQ emissions data to five, four, or three significant digits, the simulation runtime on a non-dedicated environment was slightly reduced, disk space usage was substantially reduced, and model accuracy remained relatively unchanged compared to the base CMAQ simulation, which suggests that the precision of the emissions data could be reduced to more efficiently use computing resources while minimizing the impact on CMAQ simulations.

## 1. Introduction

The Community Multiscale Air Quality (CMAQ) model (Byun and Schere, 2006) is a sophisticated three-dimensional Eulerian (gridded) numerical modelling system that uses scientific first principles to simulate the chemical transformation and transport of ozone, particulate matter, toxic compounds, and acid deposition. Since the formation and transformation of chemical species are functions of complex atmospheric and chemical interactions, two primary input types are required to initialize CMAQ simulations: meteorology and emissions. First, meteorological data (such as temperature, wind, cloud formation, and precipitation rate) provide atmospheric conditions to drive CMAQ. The second required input field, which is the focal point of this study, is emissions data (i.e., emission rates from emission sources) that characterize pollutants from both man-made and naturally occurring sources.

The chemical transport model within CMAQ typically requires multiple emission datasets which occupy a significant amount of disk space. Although disk space is becoming progressively cheaper and more affordable, the research and computational needs are rapidly increasing and becoming more complex. For instance, the total sizes of emission and meteorological datasets are about 7.0 GB and 6.8 GB, respectively for a one-day CMAQ simulation for the contiguous U.S. with a horizontal resolution of 12 km. The total disk space size for one day of output is 20 GB (for a typical output configuration considering only surface output and neglecting extra diagnostic output). Including 3D fields and diagnostic output, however, the total output disk space size can easily be tripled. Most studies with CMAQ on this scale create at least a full year's worth of data, so aggressive disk space management approaches could be justified to minimize overall costs associated with running CMAQ. Aggressive disk





space management could be a substantial cost-savings measure, regardless of whether simulations are conducted onsite (such as with a high-performance computing architecture or a Linux cluster) or by using cloud computing, where data retrievals can quickly elevate costs. Here, we propose optimizing disk space by compressing CMAQ emission datasets as one practical

consideration to maximize storage capacity. If successful, this option could be extended to other input types with large disk space needs, such as meteorological data.

Compression algorithms can be described as either lossless or lossy. Lossless compression algorithms reduce disk space by replacing repeated sequences with a smaller, unique identifier. Thus, an entire dataset can be retrieved, once uncompressed, without alteration of the original dataset (hence the name, lossless). Lossy algorithms, however, in terms of numeric arrays,

reduce disk space by manipulating the mantissa of a single floating point. Typically, trailing, or insignificant bits, are replaced with a sequence of zeros or ones. As a result, data is compressed at the cost of numeric inconsistencies between the original dataset and the compressed dataset.

The concept of maximizing disk space by altering netCDF datasets has been examined previously by Zender (2016) and Kouznetsov (2020). Zender (2016) created a versatile toolset that compresses data based on user specifications that are applied

to the mantissa of floating-point datasets. The first notable algorithm developed by Zender (2016) is precision-trimming, which is publicly available in the netCDF operators (NCO, http://nco.sourceforge.net/nco.html) utility. Precision-trimming sets all non-significant bits to zero (bit shaving) which, based on analysis, produces an undesirable bias of the compressed data (Zender 2016). As a result, Zender (2016) introduced a Bit Grooming algorithm (default algorithm in NCO) that shaves (to zero) and sets (to one) the least significant bits of consecutive values. Despite the additional toolset, Kouznetsov (2021) found substantial

artifacts, or numeric inconsistencies, in multipoint statistics caused by Bit Grooming. Due to the suboptimal results, Kouznetsov (2021) developed and evaluated multiple lossy compression algorithms with respect to NCO's available toolsets from Zender (2016). Kouznetsov (2021) created a round and halfshave lossy compression algorithm which both doubled compression accuracy by rounding the mantissa to the nearest value that has 0 tail bits and by setting all tail bits to zero except for the most significant bit which gets set to one (Kouznetsov, 2021).

Excluding analyses conducted on datasets via lossy compression algorithms, the authors are unaware of any studies that have been conducted on the compression efficiency on floating-point datasets with respect to $n$ significant digits. Additionally, Zender (2016) and Kouznetsov (2021) did not conduct evaluations regarding the impact of altered datasets on numeric simulations. In this study, netCDF datasets will be altered and compressed to explore compression efficiency, and the resultant altered datasets will be used to run CMAQ simulations to quantify the impacts on runtime and on model accuracy as a result

of dataset manipulation via a lossy compression algorithm. This study proceeds as such: in Sect. 2, a description of the methodology will be provided, followed by results in Sect. 3, then the conclusions in Sect. 4.



## 2. Methodology

All input and output files in this study are 32-bit, binary, netCDF files which inherently contain seven or eight significant digits at most. To perform this study, we created a simple tool written in Fortran to truncate floating-point data in netCDF files by keeping $n$ significant digits which are normalized in scientific notation. Table 1 shows several examples of this numerical manipulation.

We applied this tool to alter the precision of two different datasets (input emission and CMAQ model output) by keeping $n$ significant digits. For this study, CMAQ v5.3.1 (USEPA 2019) using 12 km grid spacing. We conducted four annual CMAQ simulations for 2016: one with unaltered emission data (simulation *orig*) and three with altered emissions data by setting $n$ to five (*A05*), four (*A04*), and three (*A03*) for all emission input files (gridded_no_rwc, gridded_rwc, ptnonipm, ptegu, ptagfire, ptfire, ptfire_othna, pt_oilgas, cmv_c3_12, cmv_c1c2_12, and othpt) utilized by CMAQ for this study. On the output side, direct CMAQ output (ACONC, APMDIAG, DRYDEP, and WETDEP1) from the *A05*, *A04*, and *A03* (in which *A0n* signifies an altered simulation which utilized altered emissions data to $n$ significant digits) simulations was similarly altered to possess five, four, or three significant digits (denoted as *FX05*, *FX04*, and *FX03,* respectively in which *FX0n* signifies an altered case which was post-processed by an *A0n* simulation's CMAQ output). Emission input and CMAQ output data were compressed by gzip (GNU Gzip, https://www.gnu.org/software/gzip) and bzip2 (https://www.sourceware.org/bzip2) for all simulations and cases to determine compression efficiency in terms of the reduction of disk space. In summary, there are four separate simulations (called *orig* or abbreviated as *A0n*) and nine additional, altered output cases (abbreviated as *FX0n*). For example, a CMAQ simulation that was run with emissions data that was processed with $n$ equals five significant digits, then post-processed to possess three significant digits, is denoted as *A05FX03* (see Table 2 for more simulations and cases).

Simulated numeric, or predictive, accuracy was analyzed against concentrations of particulate matter with diameter less than 2.5 μm ($PM_{2.5}$), ozone ($O_3$), ammonia ($NH_3$), the wet deposition rates of sodium (Na), ammonium ($NH_4$), chlorine (Cl), nitrate ($NO_3$), sulfate ($SO_4$), and the dry deposition rate of $O_3$ for all simulations and cases. $PM_{2.5}$ and $O_3$ were evaluated at in situ stations from the United States Environmental Protection Agency's (U.S. EPA) Air Quality System (AQS; Fig. 1.b) dataset. $NH_3$ was evaluated at in situ stations utilizing observations from the Ammonia Monitoring Network (AMON; Fig. 1.c). Hourly observations of $O_3$ were processed to calculate the maximum 8-hour daily average concentrations (MDA8) and paired in space and time with calculated MDA8 $O_3$ from post-processed CMAQ output. Likewise, daily averaged $NH_3$ and $PM_{2.5}$ observations were used to evaluate CMAQ. Observed values are paired with the volume-average pollutant estimate from CMAQ's surface layer's grid cell containing the air quality monitoring site (i.e., nearest neighbor). Statistical metrics were also calculated by pairing gridded values from the *orig* simulation (considered observed values) and the altered simulations and cases (considered the predicted values). Tabulated statistical metrics for grid−grid pairing was computed by taking the mean of specific hourly, statistical metrics.

Typical statistical metrics including mean bias (MB), correlation coefficient (r), root-mean-square-error (RMSE) and normalized mean bias (NMB) are used to evaluate all chemical species in this analysis at different temporal intervals and for





different pairing methodologies (either grid-point or grid-grid) which includes regional stratification (based on regions from

Fig. 1.a) for several figures. The utilized statistical metrics are denoted below in Eq. (1) through Eq. (4).

$$\text{MB} = \frac{1}{N} \cdot \sum_{i=1}^{N}(Y_i - X_i) \,, \tag{1}$$

$$\text{r} = \frac{1}{N-1} \cdot \frac{\sum_{i=1}^{N}((X_i - \bar{X}) \cdot (Y_i - \bar{Y}))}{\sigma_X \cdot \sigma_Y} \,, \tag{2}$$

$$\text{RMSE} = \sqrt{\frac{\sum_{i=1}^{N}(Y_i - X_i)^2}{N}} \,, \tag{3}$$

$$\text{NMB} = \frac{\sum_{i=1}^{N}(Y_i - X_i)}{\sum_{i=1}^{N}(X_i)} \cdot 100 \,\% \,, \tag{4}$$

Where $N$ is the total number of observed and predicted pairs, $X$ is the observed value, $Y$ is predicted value, σ is the standard

deviation of a distribution, and the overbars in Eq. (2) refers to the mean of a distribution.

Although many compression toolsets exist and optimization is dependent on multiple factors (Kryukov et al., 2020), gzip and

bzip2 are the most public, reliable, and widely used compressors. Both utilities are lossless compression algorithms which are

available for Linux users. In terms of functionality, gzip uses a compression algorithm called, Deflate (Deutsch, 1996) which

reduces sequences of datasets by incorporating a combination of LZ77 dictionary coding (Ziv and Lempel, 1977) and Huffman

entropy coding (Huffman, 1952). In comparison, bzip2 uses the Burrows-Wheeler (Burrows and Wheeler, 1994) algorithm

which chronologically reduces sequences of datasets by processing sequences through multiple layers of compression

algorithms. In terms of compression ratio, bzip2 is notably better than gzip, however, with respect to compression speed, gzip

is significantly faster than bzip2. Due to their availability and efficiency, both gzip and bzip2 are utilized in this study (default

settings).

**Table 1: Multiple example transformations of floating points from their original forms (first column) to their altered forms (second to fourth column).**

| Original (orig) | Altered 5 (A05) | Altered 4 (A04) | Altered 3 (A03) |
|---|---|---|---|
| 0.005666635 | 0.0056666 | 0.005667 | 0.00567 |
| $3.437405 \times 10^{-6}$ | $3.4374 \times 10^{-6}$ | $3.437 \times 10^{-6}$ | $3.44 \times 10^{-6}$ |
| 0.0005319762 | 0.00053198 | 0.000532 | 0.000532 |
| $3.437 \times 10^{-6}$ | $3.437 \times 10^{-6}$ | $3.437 \times 10^{-6}$ | $3.44 \times 10^{-6}$ |

**Table 2: Setup of all simulations (*orig*, A05, A04, and A03) and cases analyzed in this study.**

| Unaltered Emissions Data | Altered Emissions Data |
|---|---|

| a) | Simulation: *orig* | b) | Simulation: *A05* | c) | Simulation: *A04* | d) | Simulation: *A03* |
|---|---|---|---|---|---|---|---|

**Altered CMAQ Output**

| | | e) | Case: *A05FX05* | h) | Case: *A04FX05* | k) | Case: *A03FX05* |
|---|---|---|---|---|---|---|---|
| | | f) | Case: *A05FX04* | i) | Case: *A04FX04* | l) | Case: *A03FX04* |
| | | g) | Case: *A05FX03* | j) | Case: *A04FX03* | m) | Case: *A03FX03* |



**Figure 1: Regions for spatial and temporal stratification (a), AQS stations (b), and AMON stations (c) for the proceeding evaluation.**



**3.    Results**

**3.1. Data Storage**

CMAQ input and output data are stored for future analyses and to ensure the reproducibility of modeling studies which demands a tremendous amount of disk space for input and output files. Therefore, we propose to ease the disk space burden by utilizing efficient compression algorithms. For this section of the analysis, two popular, reliable, and efficient compression

utilities, gzip and bzip2, were utilized to determine compression efficiency with respect to emission input (emissions mentioned in section 2.) files and CMAQ output (mentioned in section 2. including CGRID, CONC, and SOILOUT) files. Both compression utilities were applied daily to compress emission input and CMAQ output files throughout the entirety of 2016 (Fig. 2).

The gzip compression utility reduced the file sizes by an average of 1 %, 5 %, and 21 %, or about 5 GB, 26 GB, and 111 GB,

relative reduction in file size of the compressed daily *A05*, *A04*, and *A03* emissions datasets for 2016, respectively, compared to the *orig* case. The reduction in file size (using gzip) was more substantial when applied to direct CMAQ output, with an average reduction in file size of 4 %, 19 %, and 47 %, or about 167 GB, 839 GB, and 2016 GB for *A05*, *A04*, and *A03*, respectively. With the bzip2 utility, the reduction in magnitude is much larger than with gzip, with an average reduction of file size equal to 6 %, 25 %, and 48 %, or 27 GB, 126 GB, and 241 GB, respectively for *A05*, *A04*, and *A03* emissions files and 19

%, 47 %, and 69 %, or 856 GB, 2142 GB, and 3115 GB, respectively, for the compressed CMAQ output. Thus, bzip2 is found to be a more effective tool than gzip by roughly 5 %, 20 %, and 27 % for emission data and 15 %, 28 % and 23 % for CMAQ output, for *A05*, *A04*, and *A03* (preprocessed and postprocessed data), respectively.



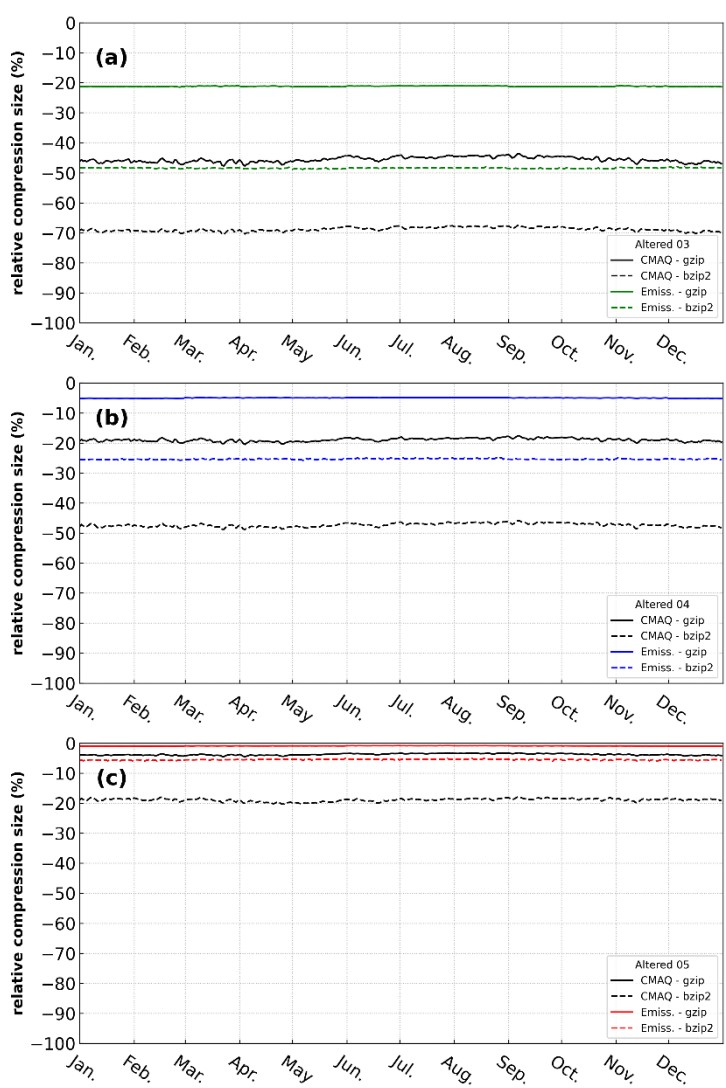

**Figure 2: Relative compression size of two utilities, gzip (solid line) and bzip2 (dotted line), on daily emission files (labelled as Emiss.)**
**and direct CMAQ output (labelled as CMAQ) for 2016 with ndigit (via the *FX* program) set to 5, 4, and 3 (labelled as Altered 05,**
**Altered 04, and Altered 03, respectively). Negative values indicate better compression efficiency.**

### 3.2. Runtime

We examined daily runtime for CMAQ using emissions data prepared with truncations of *A05*, *A04*, and *A03* compared with running CMAQ with unaltered (*orig*) emissions data (Fig. 3). Even though the simulations were not performed in a dedicated

environment (results are not entirely consistent due to the allocation of resources when the simulations were initialized), the daily runtimes for *A05*, *A04*, and *A03* were consistently lower than the runtime of the *orig* simulation. The total runtimes for



the *A05, A04,* and *A03* simulations were 3.13, 2.94, and 12.84 hours faster than the *orig* case (2 %, 2 %, and 7 %, respectively of relative reduction of runtime).

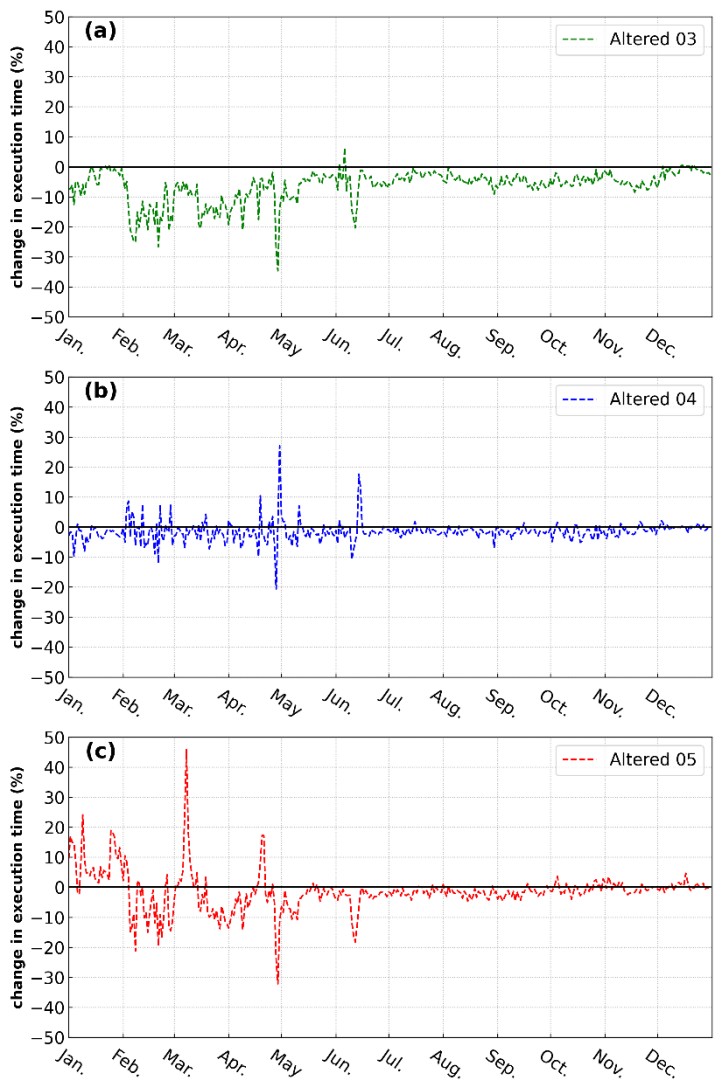

**Figure 3: Relative daily run time with respect to different adjusted emission input for the A03, A04, and A05 simulations for 2016.**

### 3.3. Accuracy

The accuracy of each case is first examined grid-to-point between modeled output and in situ observations (Fig. 1; AQS and AMON) for all available modeled pairs throughout 2016. Resultantly, each case performed with negligible differences when compared against the *orig* case in terms of statistical parameters provided for daily $PM_{2.5}$, MDA8 $O_3$, and daily $NH_3$ (Tables

3−5). For example, the bulk statistic range of NMB for daily $PM_{2.5}$, MDA8 $O_3$, and daily $NH_3$ amongst all cases is 1.845



$\times 10^{-3}$ %, $1.088 \times 10^{-4}$ %, and $7.873 \times 10^{-4}$ %. Therefore, at most, simulations deviated from one another by $1.845 \times 10^{-3}$ % (for PM$_{2.5}$) utilizing bulk statistical metrics of NMB. Similarly, the range for all statistical metrics (Table 3−5) are quite small.

Bulk statistical results (Tables 3−5) are encouraging; differences are small ignoring regional or temporal stratification. To
determine if statistical results fluctuate spatially (by region) and or temporally (by season), RMSE was computed for 9 different sub-regions (regions are portrayed in Fig. 1) across the United States for four seasons (Winter, Spring, Summer, and Fall) from the mentioned observation and model pairs. Each region's RMSE was stacked together, by simulation and case, and plotted as 'accumulated RMSE' by species. Likewise, results have negligible differences for daily PM$_{2.5}$, MDA8 O$_3$, and daily NH$_3$, respectively (Fig. 4) for all regional and temporal stratifications and for all simulations and cases.

Results indicate that all simulations and cases have negligible differences in terms of bulk statistical metrics across the U.S. and considering regional and temporal stratifications. To identify small scale, temporal discrepancies, absolute bias ($|Y_i - X_i|$) was calculated for all daily PM$_{2.5}$, MDA8 O$_3$, and daily NH$_3$ observation and model pairs for all simulations and cases. Similarly, results suggest little impact in terms of range of maximum and minimum absolute bias. In fact, the ranges of maximum absolute bias for daily PM$_{2.5}$, MDA8 O$_3$, and daily NH$_3$, computed from the biases of each simulation and case, are
$0.16 \, \mu g \, m^{-3}$, $0.13$ ppb, and $0.001 \, \mu g \, m^{-3}$ respectively (Table 6). In contrast, the 24-hour fine particle limit for PM$_{2.5}$ exposure is $35 \, \mu g \, m^{-3}$ and the 8-hour exposure limit for O$_3$ is 70 ppb (U.S. EPA Criteria Pollutants, https://www.epa.gov/criteria-air-pollutants/naaqs-table). Thus, the ranges of maximum absolute bias for PM$_{2.5}$ and O$_3$ do not exceed air quality criteria ($0.16 \, \mu g \, m^{-3} < 35 \, \mu g \, m^{-3}$ and $0.13$ ppb $< 70$ ppb) which suggests the differences in simulations and cases are negligible.

Statistical results conducted on in situ observations were redone (methodologically) at the grid level for hourly PM$_{2.5}$, O$_3$, and
NH$_3$, using the *orig* simulation (as the observed field) with respect to the altered simulations and cases (predicted fields). RMSE was first calculated for all hourly grid−grid pairs for PM$_{2.5}$, O$_3$, and NH$_3$. Next, the average, hourly RMSE was calculated for each season and region based on spatial and temporal masking using the regions portrayed in Fig. 1.a. All stratifications were grouped together as accumulative, stacked bar plots for different seasons by simulation or case. Although differences are evident (Fig. 5), the scale of such differences is quite small. For example, the total accumulative RMSE for
PM$_{2.5}$, O$_3$, and NH$_3$ (sum of all region's RMSE) did not exceed $0.1 \, \mu g \, m^{-3}$, $0.4$ ppbV, and $0.1$ ppbV, respectively for all cases and for all seasons.

Additionally, the maximum absolute bias for all grid cells was determined spatially between the *orig* simulation and the altered simulations and cases throughout 2016 for PM$_{2.5}$, O$_3$, and NH$_3$ from gridded, hourly (CMAQ) output. For PM$_{2.5}$, all simulations and cases performed similarly in which no visual differences are apparent (Fig. 6). For O$_3$ (Fig. 7) and NH$_3$ (Fig. 8), however,
the differences become relatively large for cases $n = 3$. In fact, for both species, spatial and magnitude error visibly increases with fewer significant digits (simulations and cases). For example, maximum absolute bias is largest for the *A03* simulations and even worse for the *FX03* altered cases ignoring the artifact of error across the Northeast U.S. for O$_3$ for the *A05* simulations and cases (induced by the *A05* simulation). The maximum absolute bias ranges, found by taking the range of all altered cases,





for PM$_{2.5}$, O$_3$, and NH$_3$ are 46.77 µg m$^{-3}$, 0.4265 ppbV, and 18.78 ppbV, respectively (Table 7). The minimum absolute bias
ranges for PM$_{2.5}$, O$_3$, and NH$_3$ are 5.573 µg m$^{-3}$, 0.5091 ppbV, 9.778 ppbV (Table 7), respectively. Based on range, error can
potentially be quite large compared to the statistics provided in Fig. 5, however, large-scale error is not persistent based on the
small accumulated RMSE for all regions grouped by CMAQ simulation and case (Fig. 5. Y-limits did not exceed 0.1
µg m$^{-3}$,0.4 ppbV, and 0.1 ppbV for PM$_{2.5}$, O$_3$, and NH$_3$ respectively) and by predominately low magnitude, maximum absolute
bias in Fig. 6–8. Therefore, significant error is associated with brief spikes of certain species within and around source regions.
The final aspect of this evaluation explores aspects of important species using boxplots (minimum, 25$^{th}$ percentile, median,
75$^{th}$ percentile, and maximum) of hourly, spatially averaged r, BIAS, NMB, and RMSE for all grid-to-grid pairs for deposition
rates with respect to the *orig* case. Boxplots were created for the wet deposition rates of sodium (Na), ammonium (NH$_4$),
chlorine (Cl), nitrate (NO$_3$), sulfate (SO$_4$), and the dry deposition rate of O$_3$ for all altered simulations and cases. Overall, all
species and cases perform similarly with respect to the *orig* case, and hence, amongst each other (Fig. 9). Additionally, *FX03*
performed worst and *FX05* performed best with respect to the *orig* case (in terms of similarity).

**Table 3: Annual bulk statistical metrics for all grid-point pairs for daily averaged PM$_{2.5}$ (µg m$^{-3}$) binned by simulation or case (row).**

| Case | Bias | NMB (%) | r | RMSE |
|---|---|---|---|---|
| orig | -0.02828948 | -0.37369379 | 0.53041275 | 5.01579136 |
| A05FX05 | -0.02826335 | -0.37334858 | 0.53041176 | 5.01580798 |
| A05FX04 | -0.02825803 | -0.37327833 | 0.53041128 | 5.01581581 |
| A05FX03 | -0.02815542 | -0.37192291 | 0.53041549 | 5.01582310 |
| A05 | -0.02826358 | -0.37335166 | 0.53041173 | 5.01580801 |
| A04FX05 | -0.02828796 | -0.37367363 | 0.53041358 | 5.01578126 |
| A04FX04 | -0.02828248 | -0.37360124 | 0.53041339 | 5.01578634 |
| A04FX03 | -0.02817979 | -0.37224478 | 0.53041693 | 5.01580237 |
| A04 | -0.02828831 | -0.37367825 | 0.53041352 | 5.01578145 |
| A03FX05 | -0.02829498 | -0.37376643 | 0.53041301 | 5.01577697 |
| A03FX04 | -0.02828955 | -0.37369463 | 0.53041289 | 5.01578215 |
| A03FX03 | -0.02818791 | -0.37235202 | 0.53041394 | 5.01582105 |
| A03 | -0.02829509 | -0.37376787 | 0.53041302 | 5.01577689 |
| Range | $1.397 \times 10^{-4}$ | $1.845 \times 10^{-3}$ | $5.650 \times 10^{-6}$ | $4.621 \times 10^{-5}$ |

**Table 4: Annual bulk statistical metrics for all grid-point pairs for MAD8 O$_3$ (ppb) binned by simulation or case (row).**

| Case | Bias | NMB (%) | r | RMSE |
|---|---|---|---|---|
| orig | -1.70888518 | -4.07590175 | 0.76393761 | 7.93497772 |
| A05FX05 | -1.70888101 | -4.07589180 | 0.76393721 | 7.93498192 |
| A05FX04 | -1.70888319 | -4.07589700 | 0.76393702 | 7.93498491 |
| A05FX03 | -1.70884936 | -4.07581631 | 0.76393705 | 7.93497690 |



| | | | | |
|---|---|---|---|---|
| **A05** | -1.70888073 | -4.07589112 | 0.76393725 | 7.93498117 |
| **A04FX05** | -1.70888893 | -4.07591070 | 0.76393754 | 7.93497943 |
| **A04FX04** | -1.70889025 | -4.07591383 | 0.76393745 | 7.93498084 |
| **A04FX03** | -1.70885773 | -4.07583627 | 0.76393726 | 7.93497642 |
| **A04** | -1.70888837 | -4.07590935 | 0.76393758 | 7.93497873 |
| **A03FX05** | -1.70889237 | -4.07591889 | 0.76393656 | 7.93499351 |
| **A03FX04** | -1.70889496 | -4.07592507 | 0.76393625 | 7.93499862 |
| **A03FX03** | -1.70886892 | -4.07586296 | 0.76393549 | 7.93500341 |
| **A03** | -1.70889202 | -4.07591806 | 0.76393658 | 7.93499319 |
| **Range** | $4.560 \times 10^{-5}$ | $1.088 \times 10^{-4}$ | $2.120 \times 10^{-6}$ | $2.699 \times 10^{-5}$ |


**Table 5: Annual bulk statistical metrics for all grid-point pairs for daily averaged NH$_3$ ($\mu$g m$^{-3}$) binned by simulation or case (row).**

| Case | Bias | NMB (%) | r | RMSE |
|---|---|---|---|---|
| **orig** | -0.42796669 | -35.05920328 | 0.51400293 | 1.28576807 |
| **A05FX05** | -0.42796733 | -35.05925543 | 0.51400262 | 1.28576885 |
| **A05FX04** | -0.42796814 | -35.0593213 | 0.51400369 | 1.28576660 |
| **A05FX03** | -0.42795978 | -35.05863723 | 0.51399115 | 1.28580217 |
| **A05** | -0.42796762 | -35.05927875 | 0.51400259 | 1.28576893 |
| **A04FX05** | -0.42796643 | -35.0591816 | 0.51400300 | 1.28576777 |
| **A04FX04** | -0.42796650 | -35.05918764 | 0.51400269 | 1.28577030 |
| **A04FX03** | -0.42796002 | -35.05865625 | 0.51399133 | 1.28579903 |
| **A04** | -0.42796635 | -35.05917526 | 0.51400305 | 1.28576768 |
| **A03FX05** | -0.42796899 | -35.0593916 | 0.51399920 | 1.28577519 |
| **A03FX04** | -0.42796939 | -35.05942449 | 0.51400001 | 1.28577173 |
| **A03FX03** | -0.42796703 | -35.05923092 | 0.51398972 | 1.28579406 |
| **A03** | -0.42796888 | -35.05938233 | 0.51399862 | 1.28577635 |
| **Range** | $9.610 \times 10^{-6}$ | $7.873 \times 10^{-4}$ | $1.397 \times 10^{-5}$ | $3.557 \times 10^{-5}$ |





**Figure 4: Stacked bar plots of RMSE (y) stratified by region (color), simulation and case (x), and season (subplot) for daily PM$_{2.5}$, MDA8 O$_3$, and daily NH$_3$ calculated from in situ observation.**

**Table 6: Maximum and minimum biases (modeled – observation) for all simulations and cases (row) by variable (column) throughout 2016 utilizing in situ observations.**

| Case | PM$_{2.5}$ (µg m$^{-3}$) | | Ozone (ppb) | | Ammonia (µg m$^{-3}$) | |
|---|---|---|---|---|---|---|
| | Max. | Min. | Max. | Min. | Max. | Min. |
| **orig** | 511.6467 | -161.4475 | 139.835 | -80.685 | 11.3 | -7.64904 |
| **A05FX05** | 511.6467 | -161.4475 | 139.835 | -80.685 | 11.3 | -7.64904 |
| **A05FX04** | 511.6567 | -161.4475 | 139.825 | -80.68 | 11.3 | -7.64903 |
| **A05FX03** | 511.5867 | -161.4457 | 139.745 | -80.682 | 11.301 | -7.64906 |





| | | | | | | |
|---|---|---|---|---|---|---|
| **A05** | 511.6467 | -161.4475 | 139.835 | -80.685 | 11.3 | -7.64904 |
| **A04FX05** | 511.6367 | -161.4475 | 139.845 | -80.685 | 11.3 | -7.64904 |
| **A04FX04** | 511.6367 | -161.4475 | 139.855 | -80.685 | 11.301 | -7.64904 |
| **A04FX03** | 511.5867 | -161.4457 | 139.745 | -80.682 | 11.301 | -7.64905 |
| **A04** | 511.6367 | -161.4475 | 139.845 | -80.685 | 11.3 | -7.64904 |
| **A03FX05** | 511.7467 | -161.4474 | 139.845 | -80.684 | 11.3 | -7.64908 |
| **A03FX04** | 511.7467 | -161.4474 | 139.855 | -80.683 | 11.3 | -7.64908 |
| **A03FX03** | 511.7367 | -161.4455 | 139.875 | -80.682 | 11.3 | -7.64912 |
| **A03** | 511.7467 | -161.4474 | 139.845 | -80.684 | 11.3 | -7.64908 |
| **Range** | 0.16 | 0.002 | 0.13 | 0.005 | 0.001 | $9.00 \times 10^{-5}$ |





**Figure 5: Stacked bar plots of RMSE (y) stratified by region (color), simulation and case (x), and season (subplot) for hourly PM₂.₅, O₃, and NH₃ calculated from grid–grid pairs with respect to the *orig* simulation.**





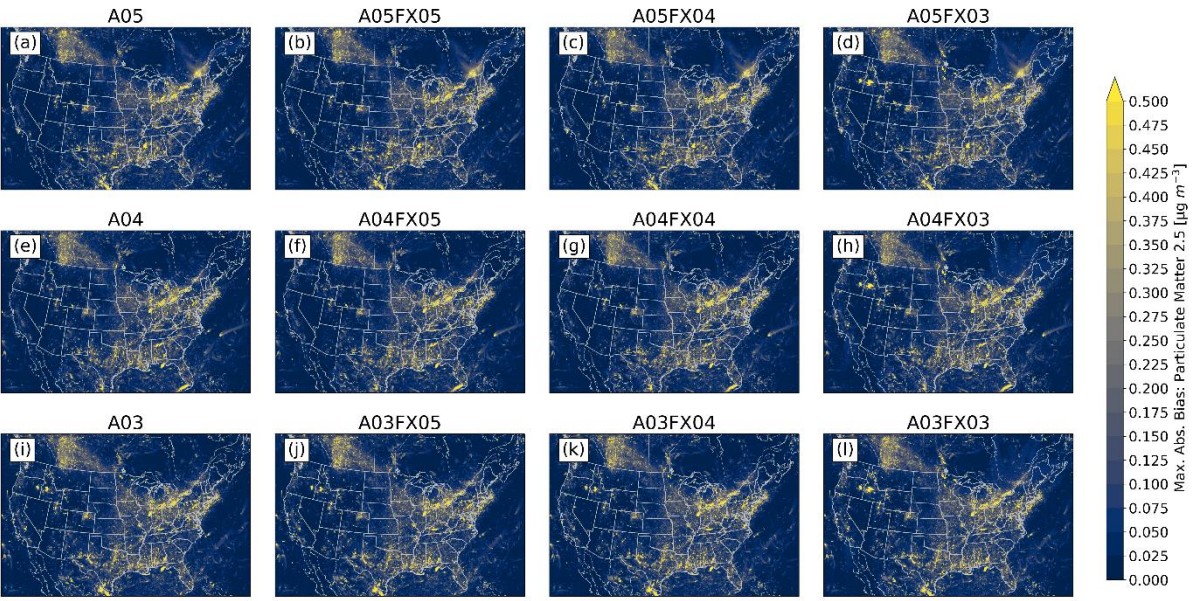

**Figure 6: Maximum absolute bias (versus the *orig* simulation) for PM$_{2.5}$ calculated from hourly output for all simulations and cases.**

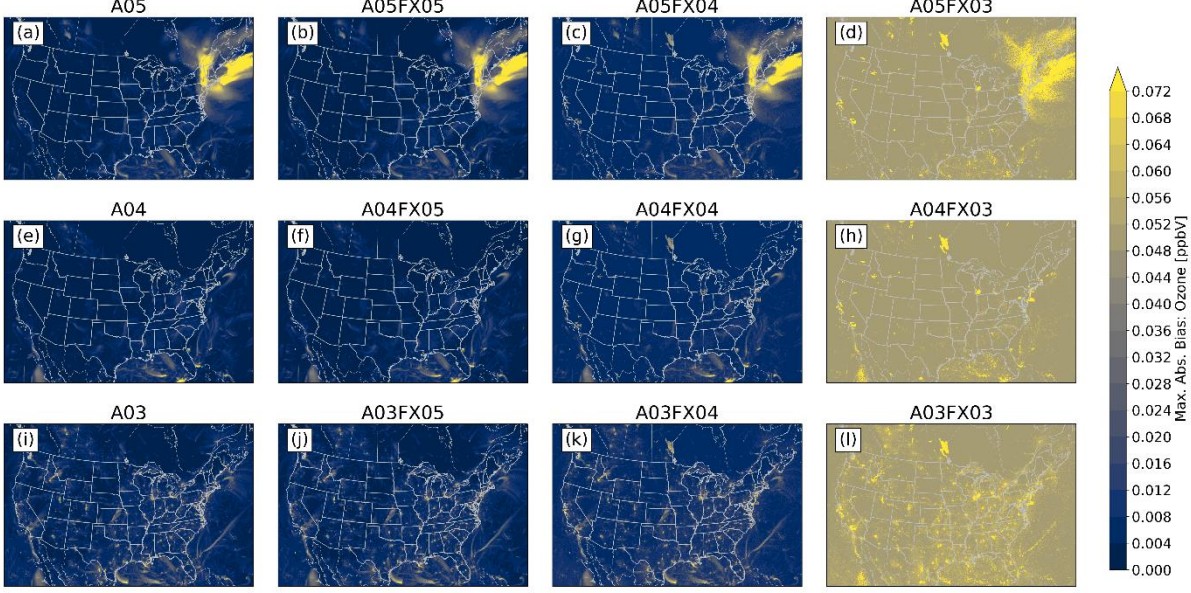

**Figure 7: Maximum absolute bias (versus the *orig* simulation) for O$_3$ calculated from hourly output for all simulations and cases.**





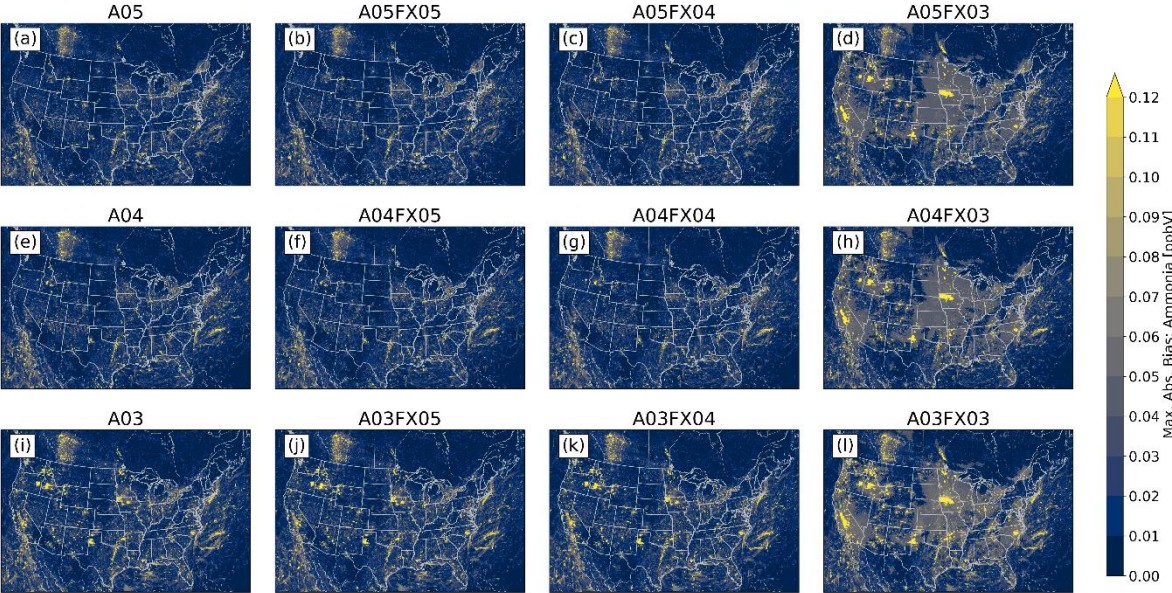

**Figure 8: Maximum absolute bias (versus the *orig* simulation) for NH₃ calculated from hourly output for all simulations and cases.**

**Table 7: Maximum and Minimum biases (*orig − altered*) calculated from hourly CMAQ output for all simulations and cases with respect to the *orig* simulation**

| Case | PM$_{2.5}$ ( µg m$^{-3}$) Max. | Min. | Ozone (ppbV) Max. | Min. | Ammonia (ppbV) Max. | Min. |
|---|---|---|---|---|---|---|
| **A05FX05** | 4.40819836 | -4.69252777 | 0.260878 | -0.08337 | 0.893507 | -0.61453 |
| **A05FX04** | 4.40777397 | -4.69240379 | 0.263882 | -0.08437 | 0.893806 | -4.01074 |
| **A05FX03** | 51.17382812 | -9.62011719 | 0.499962 | -0.50085 | 19.64355 | -4.86621 |
| **A05** | 4.40821075 | -4.69258881 | 0.260483 | -0.08343 | 0.893517 | -0.61455 |
| **A04FX05** | 4.99303246 | -4.70221233 | 0.136284 | -0.16548 | 0.875244 | -1.14815 |
| **A04FX04** | 4.99263382 | -4.70223236 | 0.136154 | -0.16448 | 1.275146 | -4.01074 |
| **A04FX03** | 51.1640625 | -9.51953125 | 0.503494 | -0.50282 | 19.64355 | -5.02832 |
| **A04** | 4.99302673 | -4.70224953 | 0.13604 | -0.16512 | 0.867432 | -1.14854 |
| **A03FX05** | 11.09228516 | -6.66992188 | 0.223785 | -0.22272 | 4.146118 | -7.44141 |
| **A03FX04** | 11.54589844 | -10.265625 | 0.225784 | -0.22272 | 4.446045 | -7.0415 |
| **A03FX03** | 41.18359375 | -9.46972656 | 0.562561 | -0.59249 | 19.64355 | -10.3923 |
| **A03** | 11.17675781 | -7.01953125 | 0.224041 | -0.22235 | 4.187866 | -7.47461 |
| **Range** | 46.76605415 | 5.57322121 | 0.426521 | 0.509121 | 18.77612 | 9.777804 |






**Figure 9: Boxplots of hourly and spatially averaged statistics (column) for multiple deposition species (row) throughout 2016.**





## 4. Conclusion

We have demonstrated that altering data by keeping a specified number of significant digits in terms of emission input and/or
simulated output, increased compression efficiency based on two different, popular compression utilities (gzip and bzip2). For
emission data, bzip2 performed far better than gzip and provided compression reduction, on average, by 6 %, 25 %, and 48 %
for emission data, and 19 %, 47 %, and 69 % for output data for the *A05*, *A04*, and *A03* cases respectively, compared to the
*orig* case. In terms of daily simulation runtime for the entire simulation year, the *A05*, *A04* and *A03* simulations consistently
faster than the *orig* simulation in a undedicated HPC system.

As for accuracy, results for all studied simulations, either with altered emission only, or with altered emission plus altered
output, produced numerically insignificant differences. For example, the bulk statistic ranges of NMB for daily $PM_{2.5}$, MDA8
$O_3$, and $NH_3$, compared to all cases and simulations at in situ locations, are $1.845 \times 10^{-3}$ %, $1.088 \times 10^{-4}$ %, and $7.873
\times 10^{-4}$ %. Similarly, small range in values is replicated for all other bulk statistical metrics such as MB, r, and RMSE. Results
stratified by region and season mimic bulk statistics. Based on the in situ evaluation, simulation performance is very similar
amongst all cases, with visible differences for the *A03* simulation and the *FX03* cases in which error is spatially detected in
Fig. 6-8.

Statistical inconsistencies arise when comparing grid−grid values of hourly $PM_{2.5}$, $O_3$, and $NH_3$ versus the *orig* simulation.
Results indicate that similarities amongst the *orig* simulation decreases with fewer significant digit simulations and cases when
analyzing the stacked and stratified (region and season) RMSE bar plot (Fig. 5). More specifically, performance with respect
to the *orig* simulation is worse for the *A03* simulation and as well, for the *FX03* cases. Such discrepancies do not occur
consistently based on results provided by boxplots of statistical metrics of deposition rates (Fig. 9). Instead, errors appear to
be confined to source regions at specific instances based on the maximum absolute (hourly) error spatial plots with respect to
the *orig* simulation (Fig. 6-8).

In summary, altering datasets by truncation to retain fewer significant digits significantly improved data compression and
slightly improved runtime. Based on the thorough, yet spatially limited, in situ evaluation, this study has shown this proposed
technique did not compromise model performance based on an evaluation of simulations and cases at in situ locations
compared to current air quality thresholds for daily $PM_{2.5}$, MDA8 $O_3$, and daily $NH_3$. These results show the benefit of altering
data by keeping five significant digits but strongly suggest keeping four significant digits and keeping three significant digits
can be considered. In addition, this proposed technique could be beneficial for groups that perform complex air quality
modeling and want to improve disk space management while negligibly impacting the quality of the simulations. Based on the
success of this study, we propose testing these techniques on the rest of CMAQ input files such as initial conditions, boundary
conditions and meteorological data to determine the viability of these techniques to more adeptly manage disk space without
compromising the quality of the CMAQ simulations used for research and to develop air quality management strategies.





## Code and data availability

The source code of the tool to alter data by keeping a specific number of significant digits and a run script which includes usage instructions, for this tool is available from DOI: 10.5281/zenodo.6620983. CMAQ 5.3.1 is available at https://www.epa.gov/cmaq/access-cmaq-source-code. Original, unaltered CMAQ input data for this study is available at https://dataverse.unc.edu/dataset.xhtml?persistentId=doi:10.15139/S3/MHNUNE. Original, unaltered CMAQ input data for this study from 1/1 – 1/5/2016 is available at DOI: 10.5281/zenodo.6624164.

## Author contribution

MW conducted the runs, performed data analysis, created graphics, and wrote the first draft of the manuscript and worked with DCW to improve it. DCW originated and oversaw this work, coded the tool to alter data by keeping a specific number of significant digits, created scripts to run the entire experiment, outlined the first draft of the manuscript, and contributed to writing and improving the manuscript.

## Competing interests

The authors declare that they have no conflict of interest.

**Disclaimer:** The views expressed in this paper are those of the authors and do not necessarily reflect the view, or policies, of the U.S. EPA.

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
