# Peer review of "The Impact of Altering Emission Data Precision on Compression Efficiency and Accuracy of Simulations of the Community Multiscale Air Quality Model"

_Geoscientific Model Development, 2022_

## Author Comment (AC1)

Referee #1

Dear Referee #1, thank you so much for reviewing this paper and providing such a thorough and constructive review and comments. Here are our responses to your comments and suggestion point by points (in blue).

GENERAL COMMENTS

Air quality models are a vital tool for air quality research and management, but these models require the use of large input data sets and they generate even larger output data sets. Moreover, the size of these data sets is expected to grow with time, so the management and archival of such large data sets is an ongoing challenge for the air quality modeling community.

This paper describes a new and useful approach to this issue: an overall "lossy" compression algorithm that reduces the size of both input and output files while preserving their important features. This is done by combining a "lossy" precision-reduction conditioning of such data sets followed by lossless data compression. The paper describes this algorithm and then assesses the costs and benefits of this new data-set management approach in terms of disk space savings, model run times, and model accuracy. Both model-to-observations and model-to-model comparisons are considered for a 2016 annual simulation with the CMAQ air quality model along with a comparison of the efficacy of two well-known lossless data compression utilities.

I found this to be a well-structured and reasonably well-written paper that would be suitable for publication in GMD. I recommend its acceptance after a number of minor revisions, including some to improve its clarity. To this end I have made a number of specific comments and suggestions below that I believe will improve the final version and that I hope the authors will consider.

SPECIFIC COMMENTS

1. The paper considers both input emissions files and CMAQ output files but the title only references input files. Should the title be expanded slightly along the lines of "The Impact of Altering Emission and Output Data Precision on ..."?

Thank you so much for the suggestion. Although we did conduct an analysis on altered-precision CMAQ output, we consider the altered-precision emission data the 'driver' for this study. Due to its significance regarding CMAQ simulations (true emphasis of this study), we'll leave the title as is.

2. Terminology matters, and the manuscript is not always clear about exactly what is being done. The word "altered" is used a lot (32 times), but using a compound modifier like "altered-precision" or "reduced-precision" instead might be clearer. The terms "pre-processed" and "post-processed" are also used (lines 24, 25, 28, 105, 110, 117, 167) in what seems to me to be a confusing way, in that "pre-processing" is used in connection with emission files, which are input files upstream of the CMAQ model, whereas "post-processing" is used for CMAQ-generated output files. However, emission files are also output files that are generated by an emissions processing system like SMOKE, and for both types of

files the precision reduction is applied after the files have been generated.  Again, referring to "reduced-precision emission files" and "reduced-precision CMAQ output files" might make it clearer that a transformation or conditioning step, namely precision reduction, has been applied to files after they have been generated.

We agree with the referee's suggestion. The manuscript has been modified accordingly. Since 'altered' was used numerous times throughout the manuscript, individual changes won't be listed here. Please examine the revised manuscript.

3. It would be helpful to the reader if a bit more detail were given in the Methodology section about (a) the characteristics of the emissions input files and CMAQ output files and (b) the 2016 base annual simulation ("orig" ):

(a) The manuscript does not give any information about the temporal resolution of the emissions files or CMAQ output files -- do they contain hourly fields or more frequent or less frequent fields?  How many different types of fields are there (e.g., different species, emissions vs. concentration vs. deposition fields)?  What is the horizontal grid size?  These details would help the reader to understand the size of the file sets.

The following sentences are added in the second paragraph of Section 2 Methodology, in the revised manuscript to address the referee's comments.

"For this study, CMAQ v5.3.1 (USEPA 2019, Appel et. al. 2021) was run with 459 columns, 299 rows, and 35 vertical layers with a horizontal grid-scale resolution of 12 km (Fig. 1.a). Emission input files consist of two area sources and nine point sources (hourly). The area source emission files contain 57 and 62 variables and the point source files contain anywhere from 54 to 58 variables (containing one vertical layer). Ten, CMAQ output files, nine of them are hourly, were generated in this study: Three output files were generated for simulation restart purposes (SOILOUT, CGRID which contains only one hour data, and MEDIA), two files contained average (APMDIAG and ACONC) and hourly (CONC) species concentrations, three files held wet deposition (WETDEP1; 140 variables), dry deposition (DRYDEP; 174 variables), and deposition velocity (DEPV; 104 variables) output, and lastly, the final file contained biogenic emission diagnostic output (B3GTS)."

(b) I know the details of the "orig" simulation are not directly relevant to the subject of this paper, but if the 'orig' simulation has already been documented in previous publications or reports, it would be helpful to have one sentence referring the reader to that additional information.  For example, Table 6 states that the maximum daily PM2.5 bias was 512 ug/m3, a very large value, which left me wondering whether the 'orig' run included wildfire emissions (line 101 does mention "ptfire", but it is not clear whether this file was used in the 'orig' run).

The following reference has been added and cited in the second paragraph in section 2 Methodology.

Appel, K. W., Bash, J. O., Fahey, K. M., Foley, K. M., Gilliam, R. C., Hogrefe, C., Hutzell, W. T., Kang, D., Mathur, R., Murphy, B. N., Napelenok, S. L., Nolte, C. G., Pleim, J. E., Pouliot, G. A., Pye, H. O. T., Ran, L., Roselle, S. J., Sarwar, G., Schwede, D. B., Sidi, F. I., Spero, T. L., and Wong, D. C.: The Community

Multiscale Air Quality (CMAQ) model versions 5.3 and 5.3.1: system updates and evaluation, Geosci. Model Dev., 14, 2867–2897, https://doi.org/10.5194/gmd-14-2867-2021, 2021.

This paper evaluated CMAQ version 5.3.1 in terms of model performance. We have adopted the same evaluation techniques from the referenced paper to determine the accuracy of our altered-precision simulations and cases. For the referenced paper (Appel et. Al, 2021), there were three kinds of fire emission input: ptfire (wildfire in US), ptagfire (agriculture related fire emission), and ptfire_othna (wildfire in neighboring countries). This is the cause for the relatively high, maximum daily PM2.5 bias (equal to 512 ug/m3).

4. References to daily measurements from the Ammonia Monitoring Network (AMoN) are incorrect. The sampling duration of the AMoN measurements is two weeks (e.g., https://www3.epa.gov/castnet/docs/AMoNfactsheet.pdf). This error should be corrected. It also raises concerns about whether the model predicted values were properly aligned with the observed values, that is, were they two-week averages. And since Tables 5 and 6 and Figure 4 present metrics for NH3 in units of µg/m3, are the units of ppbv presented in Figures 5 and 8 for NH3 correct?

We thank the referee's thoroughness regarding the sampling duration of AMoN measurements. We used AMET to pair modeled output with respect to AMoN observations in time and space. This process was done automatically, by AMET, given a set of AMET arguments. We have spoken with an AMET author, Wyat Appel, and got confirmation that AMET did pair modeled output to observations correctly according to AMoN measurement protocol.

For Table 5 and Table 6, they are model-to-observation comparisons, and the unit is µg/m3. The model values were converted prior to pairing by AMET. For Figure 5 and Figure 8, they are model-to-model comparisons. The unit for CMAQ model output is ppmv (a simple conversion to ppbv). So we believe the units are correct.

The paper has been revised as follows:

Line 127 revised manuscript: "Likewise, daily averaged PM2.5 observations and two-week averaged NH3 observations were used to evaluate CMAQ."

All mentions of "daily NH3" have been replaced with "two-week averaged NH3."

5. Table 1 is used to illustrate the output of the precision-reduction algorithm. However, it does not fully describe how rounding is handled. For example, two lines could be added to show the A05, A04, and A03 values of 100150.0 and 100250.0.

Since 100150.0 and 100250.0 are similar in terms of rounding potential, we only used 100150.0 as an additional example in Table 1.

The following block of code is the gut of the reducing-precision algorithm.

```
write( fmt, '(a2, I2.2, a1, I2.2, a1)') '(e', ndigits+7, '.', ndigits, ')'

read in input -> data
```

```
loop through c, r, and k
write(str,fmt) data(c, r, k)
read(str,*) datar(c,r,k)

write datar -> output
```

The fmt string determines the conversion of a real number into Fortran scientific notation with a specific number of digits after the decimal point. The converted number will be read in and then output to a netCDF file. Rounding follows regular standard (if the number you are rounding is followed by a number greater than 4, found the number up). The tool to reduce data precision is available at, DOI: 10.5281/zenodo.6620983, titled, "alter_data_src.tar.gz".

6. The definition of the statistical metrics on page 5, specifically on line 132, is not quite correct. The mean and standard deviations of the distribution are unknown, but they are estimated from the sample of model-measurement pairs, so the metrics are based on sample means and sample standard deviations.

Thank you for this correction. We have replaced "mean" and "standard deviation" with "sample mean of a distribution" and "sample standard deviation of a distribution", respectively.

Line 141-142 revised manuscript has been revised as "Where N is the total number of observed and predicted pairs, X is the observed value, Y is predicted value, σ is the sample standard deviation of a distribution, and the overbars in Eq. (2) refers to the sample mean of a distribution."

7. In the description of the Burrows-Wheeler algorithm, the clause "which chronologically reduces sequences of datasets by processing sequences through multiple layers of compression algorithms" (p. 5, l. 138) is not clear to me, especially the use of "chronologically". I know the Burrows-Wheeler transform is not easy to explain, but is some rewording or new wording possible?

We have revised the description of the Burrows-Wheeler algorithm in the manuscript as follows:

Line 147-148 revised manuscript: "In comparison, bzip2 uses the Burrows-Wheeler (Burrows and Wheeler, 1994) algorithm which sorts all possible rotations of an input lexically and forms an output by concatenating the last character from the sorted list."

8. There may be a discrepancy between the description of data set sizes in the Introduction (lines 57-58) and the discussion of data storage results in Section 3.1 (lines 159-165). If the CMAQ emissions files for one day are about 7 GB in size, how can the compression utilities reduce their size, again for a day, by 111 GB or 241 GB? Or if Section 3.1 is considering the emissions files for one entire year, then 7 x 365 = 2,555 GB and 21% and 48% of that number are 537 GB and 1,226 GB, respectively. Can some clarification be provided.

In lines 58 revised manuscript, it indicated a typical daily total file size of emission data, 7GB and it varies slightly each day. In lines 168-178 revised manuscript, it showed the daily relative reduction with respect to gzip and the total sum compressed file difference in the entire year of 2016. For example, the annual sum of the daily difference between compressed original emission data and compressed reduced-precision n = 3 emission data is about 111GB. The same explanation is true for bzip2 scenario. We have revised the manuscript to make it less confusing as follows:

"The gzip compression utility reduced the file sizes, on by an average byof 1 %, 5 %, and 21 %., This translates intoor about 5 GB, 26 GB, and 111 GB, actual difference between compressed orig case and relative reduction in file size of the compressed daily A05, A04, and A03 emissions datasets for the entire year of 2016, respectively, compared to the orig case. The reduction in file size (using gzip) was more substantial when applied to direct reduced-precision CMAQ output, with an average reduction in file size of 4 %, 19 %, and 647 %., This means or about 167 GB, 839 GB, and 2016 GB actual difference between orig case and for FXA05, FXA04, and FXA03, respectively for the entire year. With the bzip2 utility, the reduction in magnitude is much larger than with gzip, with an average reduction of file size equal to 6 %, 25 %, and 48 %, or  (actual differences are about 27 GB, 126 GB, and 241 GB, respectively for A05, A04, and A03 emissions files and 19 %, 47 %, and 69 % (atual differences are about, or 856 GB, 2142 GB, and 3115 GB, respectively, for the compressed CMAQ output. Thus, bzip2 is found to be a more effective tool than gzip by roughly 5 %, 20 %, and 27 % for emission data and 15 %, 28 % and 23 % for CMAQ output, for A05, A04, and A03 reduced-precision by keeping 5, 4, and 3 significant digits (preprocessed reduced-precision emission and postprocessed reduced-precision output data), respectively."

9. Section 3.2 would benefit from the addition of some discussion.  Why should reducing the precision of the input emissions result in an overall reduction in run time but also cause increases in run times for some days?  Why should there be a seasonal dependence in runtime differences, with larger differences for the first half of the annual simulation?  Why are the run-time differences larger for the A05 and A03 simulations than for the A04 simulation?  Are the decreases in overall run time expected and are they significant?  Even if these questions cannot be answered with certainty, they should at least be raised and any possible explanations offered.  The first paragraph of the Conclusion section (lines 263-264) also gives a questionable summary of Section 3.2, stating that "the A05, A04 and A03 simulations (were) consistently faster than the 'orig' simulation in a undedicated HPC system".  This statement is contradicted by Figure 3, which shows run-time increases for at least some days of all three simulations run with precision-reduced emissions.

We do not have a satisfactory explanation of the relationship between reduced-precision data and run time shown in Fig 3. It was not one of the objectives of this study, however, this interesting behavior caught our eyes. We chose to report this in the paper as an observed outcome.

All the simulations were performed in an undedicated HPC system. During the execution of each case, the model competed for I/O resources with our simulation. Hence, I/O bottleneck could be to blame for the spike in relative run-time increase. Another potential explanation is that a change in emission input could alter the pathway in aerosol dynamics calculation or reduce the number of iterations in the chemistry solver. We have revised the manuscript slightly to reflect these two possible explanations.

The following addition was made at the end of Section 3.2 Runtime: "There are two possible explanations for such behavior: First, during the execution of each case, CMAQ competed for I/O resources with other tasks on the system. As a result, an I/O bottleneck could explain spikes in relative run-time on certain simulation days (Fig. 3). Second, a change in emission input (due to the reduced-precision emission data) could alter the pathway for the aerosol dynamics calculation. This change in emission input can also reduce the number of iterations in the chemistry solver."

We used the word "consistently" to describe the overall, or average, behavior of runtime. We will replace "consistently" with "for most days" in the manuscript to avoid confusion.

10. In Section 3.3 a comparison is made between the ranges of maximum absolute bias, that is, an analysis of model errors, and the concentration values of the air quality standards (lines 202-203) but the reason for this comparison was unclear to me? Second, in the methodology description in line 206 was RMSE really first calculated for all hourly grid-grid pairs in 2016, or do you mean that grid-grid pair differences were first calculated? Otherwise, it is not clear how average hourly RMSE could then be calculated for each season and region. And when you state "all grid-grid pairs", do these include grid cells located outside the contiguous U.S.? And in lines 209-211 where you state that "total accumulative RMSE for PM2.5, O3, and NH3 (sum of all region's RMSE) did not exceed 0.1 µg/m3, 0.4 ppbV, and 0.1 ppbV, respectively for all cases and for all seasons", based on Figure 5 could these values not be smaller (perhaps 0.04 µg/m3, 0.3 ppbV, and 0.05 ppbV)?

We will address this in three parts.

Part A: It is intended to show how insignificant the differences are with respect to current air quality thresholds. Since the values are noticeably small, we will remove this line to reduce the inclusion of unnecessary information.

The following sentences were removed, "In contrast, the 24-hour fine particle limit for PM2.5 exposure is 35 µg m^(-3) and the 8-hour exposure limit for O3 is 70 ppb (U.S. EPA Criteria Pollutants, https://www.epa.gov/criteria-air-pollutants/naaqs-table). Thus, the ranges of maximum absolute bias for 24-h PM2.5 and MDA8 O3 do not exceed air quality criteria (0.16 µg m^(-3) < 35 µg m^(-3) and 0.13 ppb < 70 ppb) which suggests the differences in simulations and cases are negligible."

Part B: RMSE was calculated for each hour using all model-model pairs (at one hour) within a mask region which are illustrated in Fig. 1.a. Pairing was only done for cells within shaded regions over the contiguous US.

The following alteration was made to lines 218-219 of the revised manuscript: "Only cells that fell within each region (Fig. 1.a), within the contiguous US, were used to calculate hourly RMSE for all available regional pairs."

Part C: The values 0. 1 µg/m3 , 0.4 ppbV, and 0.1 ppbV were chosen due to the plotted maximum (Figure 9.) y-tick values.

The following alteration was done to lines 226-227 of the manuscript: "For example, the total accumulative RMSE for PM2.5, O3, and NH3 (sum of all region's RMSE) did not exceed 0.04 µg m^(-3), 0.3 ppbV, and 0.05 ppbV, respectively for all cases and for all seasons."

The following alteration was made to lines 238-240 of the manuscript: "(Fig. 5. Total accumulated values did not exceed 0.04 µg m^(-3), 0.3 ppbV, and 0.05 ppbV for PM2.5, O3, and NH3 respectively)"

11. The analysis of accuracy for hourly deposition rates at the end of Section 3.3 is helpful, but it does not directly address the main impacts of deposition, which are cumulative. Could something be said about the accuracy of seasonal or annual deposition values for the simulations and cases?

Unfortunately, we only evaluated hourly deposition rates for this study (annual). The last paragraph in Section 3.3 has been revised as:

"The final aspect of this evaluation explores aspects of important species using boxplots (minimum, 25th percentile, median, 75th percentile, and maximum) of hourly, spatially averaged r, BIAS, NMB, and RMSE for all grid-to-grid pairs (across CMAQ's domain) for deposition rates with respect to the orig case. Boxplots were created for the wet deposition rates of sodium (Na), ammonium (NH4), chlorine (Cl), nitrate (NO3), sulfate (SO4), and the dry deposition rate of O3 for all altered-precision simulations and cases. For all deposition rates, the altered-precision ndigit 3 cases (FX0503, FX0403, and FX0303) performed worst, relatively speaking, with respect to the original simulation despite negligible RMSE error differences. The altered-precision ndigit 4 cases (FX0504, FX0404, and FX0304) performed nearly identically to the ndigit 5 cases for all deposition rates excluding the wet-deposition rate of sodium and the dry deposition rate of ozone (a relatively small increase in error). The altered-precision 5 cases (FX0505, FX0405, and FX0305) and the altered simulations (A05, A04, and A03) performed nearly identically to the original simulation for all deposition rates. Interestingly, amongst the altered-precision simulations (A05, A04, and A03), A05 performed worst (very negligible error) with respect to the original simulation for all deposition rates, excluding the wet deposition rate of chlorine, when considering NMB, MB, and r. Overall, all species, simulations, and cases performed similarly with respect to the orig case, and hence, amongst each other (Fig. 9)."

TECHNICAL CORRECTIONS/SUGGESTIONS

p. 1, l. 13: Perhaps "... archive input and output data sets" or "... archive input and output files"

Done (chose the latter one).

p. 1, l. 16: "desired post-processing of the output (e.g., for evaluations or graphics)"?

Done.

p. 1, l. 21, 23: Change "losslessness compression" to "lossless compression"

Done.

p. 1, l. 22: Don't you mean "before" rather than "after"?

It should be "after". The new approach consists of two parts: 1. Reduce data precision in a file, 2. Apply lossless compression to the reduced-precision dataset. This study reports by reducing the data precision

enhances the data compression rate. In other words, reducing data precision in a file does not reduce storage but after a lossless compression utility is applied.

p. 1, l. 27:  Would this be more accurate: "To enhance the analysis of disk space efficiency, the output from the altered emissions CMAQ simulations ..."?

Done (we have also adapted the suggestion from specific comment #2).

p. 1, l. 30:  Would this be more clear: "Thus, in total, 13 gridded output products (four simulations and nine altered output cases) were ..."? (cf. line 108)

Done with the adaption from specific comment #2.

p. 3, l. 70:  Perhaps "... by manipulating the mantissa of individual floating-point numbers".

Done.

p. 3, l. 90:  "This study proceeds as follows:"

Done.

p. 4, l. 105:  "Emission input and CMAQ output data were then compressed" -- this addition would emphasize that the lossy algorithm discussed in this manuscript consists of two steps.

Agreed, but changed to: "Emission input and CMAQ output data were compressed separately …"

p. 4, l. 110:  Perhaps "...(see Table 2 for a full list of simulations and cases)"

Done.

p. 5, l. 142:  Perhaps "Examples of precision-reducing transformation of floating-point numbers from their ..."

Done.

p. 7, l. 157:  Perhaps "... throughout the entirety of the 2016 simulation"

Done.

p. 7, l. 161:  Perhaps "... when applied to reduced-precision CMAQ output files"

Done.

p. 7, l. 162:  Should this be "FX05, FX04, and FX03"?

Done. We have revised the last sentence in section 3.1 Data Storage for clarity as "… output, for *A05*, *A04*, and *A03* …" to "… output, for reduced-precision by keeping 5, 4, and 3 significant digits …"

p. 9, l. 183:  Perhaps "... all available model-measurement pairs throughout 2016".

Done.

p. 10, l. 20:  Perhaps "... maximum absolute bias for 24-h PM2.5 and MDA8 O3 do not ...".

We believe l. 20 refers to l. 203. Done.

p. 19, l. 269: Perhaps change "mimic" to "are similar to those for".

Done.

p. 19, l. 281, 285: Rather than "performance" and "quality", do you really mean "accuracy"?

Those two words have been replaced with "accuracy".

Fig. 4 and 5 captions: Perhaps expand "(x)" and "(y)" to "(x-axis)" and "(y-axis)".

Done.

Table 7 caption:  Perhaps "... with respect to the 'orig' simulation across all grid cells".

Done.

---

## Author Comment (AC2)

Referee #2

Dear Referee, #2, thank you so much for reviewing this paper and providing constructive review and comments. Here are our responses to your comments and suggestion point by points (in blue).

In this paper, the authors assess the degradation in accuracy of the CMAQ model when they decrease input/output precision to save disk space. They do a number of tests to see how the model results change when precision is degraded. For the most part, they find very little change.

We don't think the word degradation or degraded is right choice of word to describe our approach. Truncate (to 'n' significant digits) is more applicable. Certainly, by altering the data precision does not change the quality of the data or make it inferior. The accuracy of CMAQ is essentially unchanged. Our work examined a given number value in 32-bits precision how significant the last few digits are to a numerical model and if they are removed, how does it impact on data compression.

When I first got this review request, I was concerned that there would be very little to the paper. Reading a bit in, my though changed to it could provide a bit of useful information. At the end, my real concern is that they really don't get in to the most interesting results.

Let us use this opportunity to reiterate the interesting result that we offered in this work. Altering the precision of the data did not impact the model performance significantly, statistically speaking, but increases the data compression efficiency substantially by using ordinary compression tools. This is useful to most researchers who need to retain large quantities of data for a long period of time. Definitely this approach reduces storage costs significantly.

First, fundamentally what they find is that the model results change very little, on average, when they reduce emissions and stored variable precision. This is a good result, but actually, it would be of great concern if that was not the case, and would indicate that what we know about air quality models is wrong. Why do I say this? There have been lots of studies of how changes in emissions impact results, and changing the inputs by 0.1% (less on average) should have virtually no impact. Good to see that is the case. It is also expected that the change in stored variable precision would also have little impact on average. Thus, their results are very much in line with expectations.

The authors share your thoughts regarding our results. It is known that for a numerical model, when an input is perturbed, the model will produce different answers. Furthermore, researchers have done numerous emission reduction studies and the reduction amounts are in 10%, 20%, or higher range. In our study, the change is about 0.1% or less as the referee has indicated which was contributed by removing the last few digits with the 32-bits data precision. We were not aware of any studies interested in this small amount of change in emissions. We agree with the referee's comment "very much in line with expectations", however expectation is expectation. We validated such expectation with a sound scientific approach and that is what this paper is about.

An interesting twist is that they also look at run time and find little difference, though there are some spikes. Those spikes are insufficiently explored.

We shared the referee's same concern "Those spikes are insufficiently explored". We have rerun those dates with spikes and the spikes was not observed. In other words, those spikes could be considered artifacts of running jobs on a non-dedicated system and are not repeatable.

One of their most interesting results is how changes in emissions precision had a radical result on the maximum. That result begs for more analysis.

For additional analysis, we have added the following paragraph and three new figures in the supplemental document.

"Figures 6–8 show the annual maximum absolute bias of the entire domain with respect to PM2.5, O3, and NH3. We have done additional analysis by examining daily maximum absolute bias with respect to those three variables (Fig. S1–S3). By displaying the daily maximum absolute bias of the first and last day from an annual simulation: 1. regardless of simulation cases, A05, AO4, or AO3, the bias range is quite similar for all three variables, 2. daily bias coverage scale is much small compared with the equivalent annual maximum absolute bias (Fig 6-8), 3. bias from the first day did not accumulate through the entire simulation. For A03 case, NH3 daily maximum absolute bias on the first day showed some large values but the last day did not. This echoes the third point quite well. Overall, this ensures the integrity of the model performance when an altered-precision emissions were used."

A final comment is that not only did they not go far enough on both the run-time results and why the precision can have such large impacts, if even on isolated values, but answer the question, at what point do results really start to degrade?

Regarding the run time results, we concluded that run time 'spikes' were artifacts caused by simultaneously running tasks on a non-dedicated system. We determined this by rerunning the A05, A04, and A03 simulation for certain days in which a spike occurred. Upon re-running simulations, the spikes were gone or reduced in magnitude. For the FX03 case, the run time is lower than the original case most of the time; a potential explanation is that a change in emission input could alter the pathway in aerosol dynamics calculation or reduce the number of iterations in the chemistry solver.

An additional analysis was performed (please see above response). It ensures the model did not degrade when altered-precision emission was used.

Detailed comments:

Line 54: What is meant by the chemical transport model within CMAQ. Is not CMAQ a chemical transport model?

Thanks for pointing that out. CMAQ indeed is a chemical transport. To clarify that, we have revised "The chemical transport model within CMAQ" to "CMAQ model".

Ine 61: "could"? Seems a bit weak.

The phrase "management approaches could be justified" has been revised as "management is justifiable".

Line 135: remove the comma after called.

Done.

In Table 1, add a footnote to state what AOX are.Table 2 does not do its job well.  It does not say what the FX cases are.

AOX are defined on line 100 of the original version of the manuscript and on line 109 of the revised version so I don't think a footnote is necessary. FX cases are defined on line 104 of the original version of the manuscript and on line 114 of the revised version.

Line 183: "Resultantly"?  Awkward.

The word "Resultantly" has been removed.

Given the small changes, most of the tables and figures are not needed (certainly the figures are not needed as there are no real observable changes).

Despite the small differences in tabular values and geospatial images, we believe the tables and figures strongly support our conclusion. We will leave the tables and the figures in the manuscript.

---

## Author Response (AR2)

Dear Editor,

   We appreciate for your time revieing this paper and providing constructive comments and suggestions. Here are our responses point by point in blue.

Comments to the author:

1. Regarding the comments/replies to Reviewer #2:

I see none of them implemented in the most recent version (gmd-2022-82-manuscript-version6.pdf). Some are included in the author-tracked version, however not all, e.g. mentioning the additional analysis (and supplemental figures). Regarding the latter, I suggest *not* to it because 1) maximum annual biases are already a strong measure of model deviations, and 2) there is no sense comparing biases for the first and last day of the simulations for species whose lifetimes are much shorter than a year (read their budgets are exchanged by emissions/sinks several times a year). Otherwise, explicate *how* do you expect year-long biases accumulation and propagation to occur. Should you decide to keep the supplement material, please follow the rules of its preparation (title page, etc.).

A copy of the tracked version (without the tracking feature) was saved, but it wasn't used for submission due to human error. We apologize for this mistake.

The intension of showing the maximum absolute bias for $O_3$ and $PM_{2.5}$ on the first and last days of the simulations, was an attempt to resolve the 2$^{nd}$ reviewer's concern regarding (CMAQ) model degradation (does error worsen over time). We agree with your assessment about the lifespan of simulated pollutants. We will not include the supplemental figures and we will add the following text at the end of Section 3.3 in the manuscript to reassure the stability of our simulations (for reviewer 2):

"No error accumulation due to the non-systematic changes in model inputs (changing precision introduces both positive and negative changes in a spatially and temporally random manner) can occur over the course of the annual simulation for chemical species of interest such as $O_3$ and $PM_{2.5}$. Their lifetimes are much shorter than a year, i.e. their simulated budgets within the continental-scale modeling domain are repeatedly exchanged through transport, emissions, and chemical and physical sinks. All simulations (*orig*, *A05*, *A04*, and *A03*) are numerically stable (no compounding error over time)."

2. Regarding the changes to the model run time (Sect. 2.3):

Since it is not possible to disentangle the influence of external factors that have affected the run-time of the simulations, presentation of Fig. 3 is impractical. Some computer systems/codes provide alternative measures of simulation effort (e.g. cpu-time, CPU cycles or whatever metric scaling with the number of operations performed) which are independent of wall-time used. Should you possess such from your simulations, please use them in Fig. 3 and in the related discussion instead. Otherwise, Fig. 3 should be removed.

We do agree with the editor's view "Since it is not possible to disentangle the influence of external factors that have affected the run-time of the simulations, presentation of Fig. 3 is impractical.". Instead,

we believe execution time is a vital part of our study. We have provided potential explanations for the behaviour execution time which was illustrated in Figure 3. Even though these explanations might not determine the influence of external factors that have impacted simulation run-time, it will be beneficial for the community to understand that such behavior exists (being transparent and maintaining integrity).

Indeed, for n = 3, the execution time was smaller than the original case almost everyday and for the other two cases, the execution time was smaller for most days. We don't believe this is a product of random behaviour. Additionally, CMAQ is an MPI-based model (reflected in the first sentence in Section 1 in the revised manuscript). The model elapsed time (execution time depicted in Figure 3) was reported by an MPI function called, 'MPI_WTIME' (reflected in the first sentence in Section 3.2 in the revised manuscript) which utilizes a native high-resolution timer, within the model itself. We also used the Linux 'time' command which summarized real-time, user CPU time, and system CPU time spent by executing the mpirun command. The user CPU time was almost identical to the reported elapsed model time. This is a method to validate the legitimacy of the elapsed time we provided in Figure 3.

3. Presentation issues:

The values in Tables 3–6 are anything but a comprehensible way to present the changes between the orig and altered-precision simulations; the ranges between the latter have no practical meaning (or please elucidate what useful information you derive from them, I see only sense in minimum and maximum difference, but not the difference of the latter). Since you investigate the impact of altering the precision on your results, you need to present the *differences* in biases, MNB, etc. of the sensitivity simulations versus the orig case. These in turn may even provide you with a possibility of a rough extrapolation of how the accuracy changes depending on the number of reduced digits.

Thus, I advise that the bias, MNB, r, RMSE and minima/maxima from the orig simulation *only* should be presented in one table for all species (PM2.5, O3, NH3). *Changes* to the bias, RMSE and minima/maxima values should be shown in bar chart plots (one plot per characteristic, i.e. four plots maximum, please use overlapping axes for different species). Ranges should be removed. So should be the changes to r (they bear no meaning on these scales) and NMB values (they are equivalent to biases).

We have removed Tables 3 – 6 and we created Table 3 to show bulk statistical metrics of bias, NMB, r, and RMSE for the *orig* simulation so a reader can gauge the accuracy of CMAQ. To follow the editor's suggestion, we have created Figure 4 to show the absolute change in bias, RMSE, and minima/maxima with respect to $PM_{2.5}$, $O_3$, and $NH_3$. The manuscript has been revised to reflect these changes. Information about ranges, change of r, and NMB are not included.

Figures 6–8: please re-render with distinct hue colour scale (i.e. "rainbow", or any other multi-colour) and lesser number of levels – right now it is not possible to read the actual value from the colour in the plot. Same applies to Figures S1–S3, should you decide to keep these (I strongly discourage that, see the comment point 1. above).

Thank you for the suggestion. For Figures 6 - 8 (Figures 7 – 9 in the revised manuscript), we followed the guidelines < https://www.geoscientific-model-development.net/submission.html#figurestables > to make our images vision impairment friendly. We chose the color map titled, 'viridis'. Therefore, we

would like to keep them as they are.

Figure 5: please scale vertical axes according to the maxima of values presented in a given plot, or at least according to the maximum for a given species. Current scales have little sense w.r.t. to the data shown.

Figure 5 (Figure 6 in the revised manuscript) has been re-scaled.

4. Regarding the comment of Reviewer #1, Point 11.:

I very much agree with the Reviewer #1 that presenting the estimate of the changes to the *cumulative* deposition is beneficial for the study. I reckon that obtaining such is possible (you just need to integrate the hourly rates available). Therefore, please replace the analysis of hourly estimates with the integral ones (annual). Presentation of the latter should be improved similar to that described in p. 3 above. That is, integrals for orig simulation should be presented in a separate table or added to that including biases, RMSE, etc., and differences should be shown in plots.

We have calculated the sum of absolute differences (annual for all hours and cells over the US over land) for the dry deposition of ozone and for the wet deposition of sodium, ammonium, chlorine, nitrate, and sulphate throughout 2016. The values have been plotted as bar plots and are provided in Figure 10.

Beyond these four responses, we have indicated that the high bias in $PM_{2.5}$ was associated with the Pioneer wildfire in Idaho from July to September of 2016 (line 235 – 237) as well as we have updated the conclusion with an explicit recommendation: "These results show the optimal benefit of altering CMAQ input data by keeping three significant digits then subsequently keeping four significant digits for CMAQ output data."

---

## Author Response (AR3)

Dear Editor,

   Happy New Year!! We appreciate for your time reviewing this paper and all your comments and suggestions. Here are our responses point by point in blue.

Comments to the author:

1. Please remove use of "ndigit" and "FX program" (these terms are not introduced in the manuscript text) and simply refer to "FX* simulations" etc. instead.

Done.

The following text was edited in the manuscript:

"Figure 2: Relative compression size of two utilities, gzip (solid line) and bzip2 (dotted line), on daily emission files (labelled as Emiss.) and direct CMAQ output (labelled as CMAQ) for 2016 with reduced-precision settings: 5, 4, and 3 (labelled as Altered 05, Altered 04, and Altered 03, respectively). Negative values indicate better compression efficiency."

"… the altered-precision 3 cases (simulation output that was reduced to 3 significant digits; *A05FX03*, *A04FX03*, and *A03FX03*), performed equally poor, relatively speaking, with respect to the *orig* simulation. The altered-precision 4 cases (*A05FX04*, *A04FX04*, and *A03FX04*) performed nearly identically to the altered-precision 5 cases (*A05FX05*, *A04FX05*, and *A03FX05*) for all deposition rates excluding the wet-deposition rate of sodium and sulfate and the dry deposition rate of ozone. The altered-precision 5 cases (*A05FX05*, *A04FX05*, and *A03FX05*) and the altered simulations (*A05*, *A04*, and *A03*) performed nearly …"

2. Please add absolute simulated dry deposition amounts either in text or in the caption to Figure 10, otherwise one cannot judge the relative scale of changes.

Figure 10 depicts the sum of absolute difference between the base simulation (*orig*) and the altered-precision simulations and cases across the domain (over land across the contiguous US), throughout 2016, for hourly output, and with respect to the wet deposition rates of sodium, ammonium, chlorine, nitrate, and sulfate and for the dry deposition rate of ozone. We have provided the domain sum of the base case with respect to those variables, even though these sums do not reflect the Figure 10 directly, but as a simple gauge of the relative changes among all the altered-precision cases.

We have added the following in the manuscript:

"For comparison purposes, the annual sum, considering all grid cells within the contiguous U.S., for the wet deposition rates of sodium, ammonium, chlorine, nitrate, and sulfate are $1.42 \times 10^5$, $6.69 \times 10^4$, $21.75 \times 10^4$, $2.58 \times 10^5$, and $1.72 \times 10^5$ kg ha$^{-1}$, respectively for the base simulation (*orig*).

Similarly, the annual sum for the dry deposition rate of ozone (contiguous U.S.) is $2.78 \times 10^6$ kg ha$^{-1}$ for the base simulation."

3. If you desire to keep Figure 3, I understand that other system bottlenecks are difficult to estimate (i.e. yes, you can state that potentially due to lesser I/O strain your emission data loading to the model proceeds faster). Then, however, either remove the statement on l.192 or provide figures on number of iterations changes in the chemistry solver or other computing routines – this information is always available from the model and should be made available to the readers. Using statements "can also" or "we believe" is inappropriate in such cases.

Within CMAQ, there are various subroutines that utilize iterative solver such as the SOA calculation and chemistry (EBI) routine, for example. CMAQ does not, however, report the number of iterations to achieve convergence for each time step. Therefore, we regret to inform the editor that we cannot provide direct evidence to support our original statement, "This change in emission input can also reduce the number of iterations in the chemistry solver." We removed the text  from the manuscript.

Please also make sure to supply high-quality (vector) graphics during the typesetting.

We saved our images as .png and .pdf files. The .pdf files are vectorized and will be provided when applicable.

---

## Author Response (AR4)

Dear Editor,

We appreciate for your time reviewing this paper and all your comments and suggestions. Here are our responses point by point in blue.

Comments to the author:

In my reply from 10/11/2022 I have intended to request adjusting the vertical axes scales for Figure 6, however by mistake I mentioned Figure 5. Please adjust these. Figure 6 axes and caption also should explicitly mention that the *difference* to the RMSE original case is plotted (or use the Δ symbol). That is, the caption should read: "Figure 6: Stacked bar plots of changes to RMSE (y-axis) stratified by region (color), simulation and case (x-axis), and season (subplot) for hourly PM2.5, O3, and NH3 calculated from grid–grid pairs with respect to the orig simulation."

Thanks for the suggestion. We have adopted. Figure 6's caption in the revised manuscript reads as:

"Figure 6: Stacked bar plots of changes to RMSE (y-axis) stratified by region (color), simulation and case (x-axis), and season (subplot) for hourly PM2.5, O3, and NH3 calculated from grid–grid pairs with respect to the orig simulation."

Figure 5 still has unsatisfactory quality (see "Presentation issues" in my comment from 11/10/2022). Variations between data shown in each plot are impossible to judge and bear no useful information. You show the differences to the *orig* simulation in subsequent Figure 6 anyway. Please keep only *orig* data in Figure 5, and therefore: make only three plots (for PM, O3, Ammonia) in one row, and present all seasonal values in each, respectively. In summary: Only *orig* data in 3 plots, 4 stacked bars each. Manuscript text in Sect. 3.3 should be adjusted accordingly (i.e. orig biases shown in Fig. 5, differences to these in Fig. 6).

Thanks for your comments. We respectfully disagree with your assessment "Variations between data shown in each plot are impossible to judge and bear no useful information. You show the differences to the *orig* simulation in subsequent Figure 6 anyway. Please keep only *orig* data in Figure 5, …". Figure 4 and Figure 5 based on the same information, model against observations. Figure 5 shows the model performance (daily PM2.5, MDA8 O3, and two-week averaged NH3) against observations of the 12 reduced-precision cases side by side with the orig case with respect to different seasons and different regions in a stacked bar fashion. On the other hand, Figure 6 shows the grid-to-grid difference (hourly PM25, O3, and NH3) between each of the 12 reduced-precision cases and the orig case for the entire model with respect to different seasons and regions. In summary, we believe there are two reasons to keep Figure 5 in the manuscript as it is: Figure 5 shows model performance from a different angle which can be serves as an additional validation technique, from Figure 4 and Figure 5 shows a different comparison approach than Figure 6.

Please increase font sizes in Fig. 4, as during the production these will likely become unreadable.

We have increased the font size from 12 to 14.

Please go once more through the entire manuscript and carefully cross-check all table and figure numbers for consistency with the manuscript text.

Cross-checked.

---

## Author Response (AR5)

**Dear Editor,**

We appreciate for your time reviewing this paper and all your comments and suggestions. Here are our responses point by point in blue.

**Comments to the author:**

Perhaps, you did not recognise the essence of my comment regarding Figure 5. I do understand that Figure 5 and 6 show different data. I, however, cannot accept showing 12 plots each of which presents 13 \*identical\* stacked bars, this is nonsensical. You may state in the manuscript text that results for different sensitivity simulations are negligibly different. And the remaining data for \*orig\* simulation can be grouped into three plots by species, each including all seasons.

We fully recognized the essence of the Editor's comment dated 11/10/2022 "Variations between data shown in each plot are impossible to judge and bear no useful information". No doubt Figure 5 shows pretty much 13 "identical" stacked bars in all 12 different sub-plots. In this work, we reduced the precision in the emission input (the A0n cases) as well as the output (the A0nFX0m cases). You might argue that this is very similar to emission reduction sensitivity study. However, nature is guite different. We manipulated the digits in the data not changing the values in the file explicitly by a giving percentage/value. I believe we are the first one to manipulate the data in this way in the air quality modeling study. Surely researchers, regulators particularly, are very concerned about the impact of this new way of changing emission data on model performance. We followed a traditional approach to analyze data by comparing model results with observations. Figure 4 showed the model performance (12 cases with the difference between the orig case) with all available observations. We then kind of zoom in different regions as well as seasons to examine the model performance (again with all 12 cases). Thirteen "identical" stacked bars were plotted. The results showed no significant difference when our approach of manipulating the data was used. That is the exact message we want to show to the readers and eliminate their concerns. By showing Figure 5, we believe it is more powerful and convincing than the sentence "results for different sensitivity simulations are negligibly different" since this is a brandnew way of manipulating data. In addition, we are not dealing with sensitivity in the traditional way.

Perhaps, there is a plotting error in Figure 5? Or please explain me which sensible information (except that sensitivity simulations insignificantly differ from \*orig\*) I receive by looking at 13 identical stacked bars in each plot.

In this research, we proposed a new technique to reduce the precision of data (input and/or output) which improved data compression with typical tools (gzip and bzip2) substantially. The rest of this article provided proofs that reduced-precision data did not affect model performance significantly. First, we followed the traditional method by comparing model output with observations (heavily been used in the model evaluation community). We chose to examine three different components: daily PM2.5, MDA8 O3, and two-week averaged NH3, with different statistical metrics for the entire 2016. We formed the difference between 12 reduced-precision cases and the orig case and represented in Figure 4. Next, we did one step further by comparing

model output against observation (all 12 cases and the orig case) from regional and seasonal perspective and again this technique is being wildly used in model evaluation process. The result was presented in Figure 5. All 13 stacked bars were "identical" provides a powerful and convincing message that these reduced-precision cases performed very much like the orig case. We prefer the current format in Figure 5 with actual model performance, rather than showing the difference with respect to the orig case (relative sense). Furthermore, in the simulation domain, there are over 137,000 spatial grid cells and observations stations reside in a fraction of those grid cells. We then expanded our analysis by comparing all the grid cells (model to model or grid-grid) between those 12 cases and the orig case. Figure 6 gives reader a boarder sense of the model performance with our new proposed technique. These three figures seemed to show the same thing, but they provided a powerful and convincing argument that the reduced-precision cases behavioured "identical" to the orig case from different perspectives. I hope you, the Editor, can appreciate our approach of providing evidence to convince readers that our new and unique reduced-precision technique does not affect model performance significantly.